# Prenatal Co-Exposure to Manganese, Mercury, and Lead, and Neurodevelopment in Children during the First Year of Life

**DOI:** 10.3390/ijerph192013020

**Published:** 2022-10-11

**Authors:** Paulina Farías, David Hernández-Bonilla, Hortensia Moreno-Macías, Sergio Montes-López, Lourdes Schnaas, José Luis Texcalac-Sangrador, Camilo Ríos, Horacio Riojas-Rodríguez

**Affiliations:** 1Instituto Nacional de Salud Pública, Dirección de Salud Ambiental, Universidad 655, Col. Sta. Ma. Ahucatitlán, Cuernavaca 62100, Mexico; 2Unidad Iztapalapa, División de Ciencias Sociales y Humanidades, Universidad Autónoma Metropolitana, Ciudad de México 09340, Mexico; 3Unidad Académica Multidisciplinaria Reynosa-Aztlán, Universidad Autónoma de Tamaulipas, Lago de Chapala y Calle 16, Aztlán, Reynosa 88740, Mexico; 4Instituto Nacional de Perinatología, Montes Urales 800, Lomas de Virreyes, Ciudad de México 11000, Mexico; 5Instituto Nacional de Neurología y Neurocirugía, Av. Insurgentes Sur No. 3877, La Fama, Ciudad de México 14269, Mexico

**Keywords:** lead, manganese, mercury, co-exposure, prenatal, children, neurodevelopment

## Abstract

Lead (Pb), mercury (Hg), and manganese (Mn) are neurotoxic, but little is known about the neurodevelopmental effects associated with simultaneous prenatal exposure to these metals. We aimed to study the associations of Pb, Hg, and Mn prenatal levels (jointly and separately) with neurodevelopment in the first year of life. Methods: Pb, Hg, and Mn blood lead levels were measured in 253 pregnant women. Their offspring’s neurodevelopment was assessed through the Bayley Scale of Infant Development III^®^ at one, three, six, and twelve months. The metals’ mean blood levels (µg/L) were Pb = 11.2, Hg = 2.1, and Mn = 10.2. Mean language, cognitive, and motor development scores of the infants at each age were between low-average and average. Multilevel models’ results showed that language development coefficients of the offspring decreased by 1.5 points per 1 µg/dL increase in maternal blood lead levels (*p* = 0.002); the magnitude of the aforementioned association increased in children with maternal blood Mn < 9.6 µg/L (ß = −1.9, *p* = 0.003) or Hg > 1.9 µg/L (ß = −1.6, *p* = 0.013). Cognitive and motor development had negative associations with maternal blood Pb levels; the latter was statistically significant when the interaction term between Pb, Mn, and Hg was included (ß = −0.037, *p* = 0.03). Prenatal exposure to low Pb levels may impair infants’ neurodevelopment in the first year of life, even more so if they are exposed to Hg or deficient in Mn.

## 1. Introduction

Early life stages, including fetal development, have been deemed a critical window for neurodevelopment; during this time, neurodevelopmental processes are at their peak, [1] and many environmental pollutants are not blocked by the placenta [2] or the blood–brain barrier [3], hence causing structural and functional damage to the brain that can affect immediate and/or later-life health outcomes [4]. Throughout pregnancy, the placenta actively or passively transports elements that are essential, although potentially toxic at certain levels (such as Mn), and purely toxic elements (such as Hg and Pb) [5,6] and these elements can accumulate in fetal tissues such as the brain [7].

Lead (Pb) and mercury (Hg) are classified by the World Health Organization (WHO) among the ten chemicals of major public health concern [8]; these two non-essential metals are classified thus because, among other things, they are widespread environmental pollutants known to be neurotoxic at relatively low doses during early stages of development. In fact, the rate of decline in neurodevelopmental outcomes can be supralinear: greater at blood Pb levels < 5 µg/dL than at higher blood Pb levels [9,10,11], and Hg is infamously known for causing devastating neurotoxic effects to the fetus [12].

Manganese (Mn) can also be neurotoxic, but unlike Pb and Hg, it is an essential element. Therefore, both a Mn-deficient diet and excessive Mn environmental exposure may affect neurodevelopment. Studies both on humans and animals have determined that excessive Mn levels alter neurobiological systems in the basal ganglia and the dopaminergic system. However, more recent studies are showing that Mn produces effects on the cerebral cortex that may affect cognitive function at levels lower than previously recognized [13,14]. On the other hand, the required Mn concentrations in the rat are higher during fetal brain development than at any other age, suggesting that Mn requirements during infancy are elevated; in humans, a sufficient Mn supply is also known to be crucial for normal brain development [15].

Studies have usually investigated the association of neurotoxicity with metals focusing only on one of them at a time or on binary combinations. However, real-life exposures usually consist of simultaneous contact with multiple combinations of different levels of several pollutants that might produce synergistic or antagonistic effects. The chemical mixtures and their health effects vary among countries and within them; in Latin America, differences have been seen according to demographics, socioeconomic factors, and urban and industrial development [16]. For instance, positive correlations have been found between blood Pb and Mn levels in children living in urban areas [17].

Furthermore, relatively low concentrations of a metal that have not been associated with an adverse effect might be contributing to one in the presence of other neurotoxic metals. For example, an unhealthy western diet can be a source of exposure to Pb and Hg and deficient in Mn [18]. Even though Mn, Hg, and Pb may have different targets in the nervous system and different modes of action, resulting in different neurological effects, they may also share properties, such as directly altering the synaptic structure or indirectly dysregulating the levels of expression of dysregulation of neurotransmitter receptors such as glutamate, dopamine, aminobutyric acid (GABA), and acetylcholine [19,20,21].

In this study, we investigated whether concurrent in utero exposure to background Pb, Mn, and Hg was associated with language, cognitive, and motor neurodevelopment indices in boys and girls from Sonora, Mexico during their first year of life.

## 2. Materials and Methods

### 2.1. Study Design and Population

The present study is a part of a cohort study of Mexico’s National Institute of Public Health on prenatal exposure to toxicants and has previously been described in detail [22]. Briefly, we carried out this study in rural and suburban areas of Sonora in northern Mexico. From 2012 to 2015, we enrolled 386 women who were in the third trimester of pregnancy when they attended routine prenatal visits at the public health clinics of four municipalities that covered the study zone in the valleys of Yaqui and Mayo (Huatabampo, Benito Juárez, Etchojoa, and Navojoa); these clinics tend to a non-insured population of low-socioeconomic status (Figure 1).

The recruited women were followed from the third trimester of pregnancy until their offspring turned one year of age. Women–infant pairs were considered eligible for the study if they met the following criteria: the woman was between 18 and 45 years old, healthy (apparently and reportedly), pregnant with only one child, could speak, read, and write in Spanish, and the child was born after a full-term pregnancy without complications, was healthy, and weighed ≥2500 g. From the initially recruited cohort of 386 women, 133 participants were lost due to: lack of any essential information or measurement (n = 54), delivery before 37 weeks of gestation (n = 31), no longer wanting to participate (n = 23), birth weight less than 2500 g (n = 10), change of residence (n = 6), twin birth (n = 3), neonatal death (n = 3), congenital diseases (n = 2), and newborn’s Apgar score < 8 (n = 1). Our final cohort consisted of 253 mother–offspring dyads.

This study and its procedures were approved by the research, ethics, and biosecurity committees of the National Institute of Public Health, Mexico (approval number, CI: 1036).

### 2.2. Sociodemographic and Infant Development Data

#### 2.2.1. Questionnaire and Anthropometrics

The women received a thorough explanation of what their participation in the study would consist of and they read and signed an informed consent letter specifying their participation was voluntary and that they could change their mind about it with no repercussions whatsoever. They proceeded to fill out a specially designed questionnaire on their health status before and during the current pregnancy, environmental exposures, and sociodemographic characteristics.

We also assessed the maternal intelligence quotient (IQ) using the Wechsler Abbreviated Scale of Intelligence (WASI)^®^. The WASI was developed as an independent test to provide a brief and reliable measure of cognitive ability to be used in clinical, educational, and research settings. This test provides flexible administration options; the four-subtest form that was used in this study is administered in approximately 30 min. Normalization data were obtained from a large national representative sample of children and adults from 6 to 89 years [23].

#### 2.2.2. Neurodevelopment Assessment

The mother and her child were assessed in the same session by psychologists that were specifically trained for these examinations and standardized. We assessed the child’s neurodevelopment in the first, third, sixth, and twelfth months.

For the assessment of cognitive, language, and motor development of the offspring, we used the Bayley Scales of Infant Development, third edition (BSID-III)^®^ adapted for the Mexican population. BSID-III globally assesses the most important developmental areas, allowing the level of child development to be determined simply and precisely, as well as early identification of developmental delays [24,25].

The BSID-III scales provide raw scores based on chronological age to generate cognitive, language, and motor domain subscores based on US norms (ranging from 40 to 160 points). We standardized raw scores for each domain using conversion tables provided in the manual with an expected mean of 100 points and a standard deviation of 15 points [25].

### 2.3. Sample Collection

Upon recruitment, specially trained nurses drew a venous blood sample from the medial cubital vein of the pregnant women after extensive cleaning of the area with antiseptic alcohol. Blood samples were collected in EDTA purple-top Vacutainer tubes. Immediately after obtaining the blood, the tubes were inverted six times to mix the anticoagulant with the blood inside the tube, since metal determinations were done in whole blood. The samples were labeled, refrigerated at 4 °C for storage, and transported to the facilities in Mexico City where analytical determinations were done.

#### Blood Lead, Manganese, and Mercury Determination

Blood Pb was assayed by atomic absorption spectrophotometry. We used the Zeeman background correction (Spectrophotometer Perkin Elmer model AA600, autosampler AS-800). Calibration curves were constructed with a commercial standard (GFAAS mixed standard, Perkin Elmer Pure). A matrix modifier consisting of Triton X-100 (0.05%) and monobasic ammonium phosphate (0.01%), and nitric acid (0.20%) were used to dilute the standards and samples. The samples were assayed in duplicate with standard deviations less than 10% between repetitions. Quality control was assured by analyzing blood with known quantities of lead from the lead Wisconsin State Laboratory of Hygiene proficiency program. The analysis session was considered as valid if the results from the external standard were within the values reported by the program and standard deviations of repetitions were less than 10%. The results are expressed as micrograms of lead per deciliter of whole blood [26].

Manganese was determined in the electrothermal-graphite furnace equipment described above. Blood samples were diluted with a matrix modifier containing 1% ammonium-dihydrogenphosphate and 0.1% Triton X-100. Calibration curves were prepared with a commercial Mn standard. The samples were analyzed in duplicate after calibration, with a standard deviation below 10% between repetitions; if dispersion was greater than this value, samples were reanalyzed. Quality control of the analysis of blood Mn was assured by measuring biological matrix-based reference material (Bovine Liver from the National Institute of Standards and Technology, 1577b, Gaithersburg, MD, USA) along with the blood samples. The reference material was acid-digested with concentrated Suprapur nitric acid and diluted with the matrix modifier described above. The analysis was considered as valid if the results from the external standard were within the 95% confidence interval for mean Mn reported by the provider [26].

Total Hg concentrations were measured by using a cold vapor-flow injection system FIAS 100 (Perkin Elmer. Whaltham, Massachussets. USA) coupled with an atomic absorption spectrophotometer (Perkin Elmer 3110). The blood samples were digested with a mixture of acids HNO3, HClO4, and H2SO4 (5:2:1). Mercury ions in the sample were reduced using 0.2% (m/v) sodium tetrahydroborate(III) in 0.5% (m/v) sodium hydroxide in the presence of 3% (v/v) hydrochloric acid (from 30% HCl) to elemental mercury Hg. The linear range of calibration was from 1 to 20 μg/L of Hg [27]. All determinations were made in the Neurochemistry Laboratory of the Instituto Nacional de Neurología y Neurocirugía from Mexico, City.

### 2.4. Statistical Analysis

An exploratory analysis was carried out to observe the distribution of the language, motor, and cognitive BSID III and the maternal concentrations of metals. Because we did not observe a pronounced skewed distribution, the variables were not transformed. The comparison between those who completed all tests and those who did not was performed using quantiles for quantitative variables and percentages for qualitative traits. Language, motor, and cognitive BSID III scores over time were explored through box plots.

Associations between each neurodevelopment domain considering all the observations available for each child at ages of 1, 3, 6, and 12 months, and each metal’s concentration were assessed using multilevel models. Municipality of origin was the grouping variable at the second level and child ID was the grouping variable at the first level nested into Municipality. Intercepts and slopes were declared as random effects assuming that the vector of repeated measurements on each subject follows a linear regression model. Thus, each individual may have a subject-specific intercept and slope representing the different susceptibilities to the exposure among subjects. Interaction terms between each combination of two metals and all three of them were also introduced and evaluated in each model.

Language, motor, and cognitive scores were introduced in the models as discrete dependent variables. Metal concentrations were explored as continuous variables; Pb was not additionally categorized, since there is no known threshold for its prenatal neurotoxicity, but Hg and Mn were according to their statistical distribution. Since Hg is non-essential, it was dichotomized above its mean (1.9 µg/L), and Mn (an essential metal) was divided into tertiles to account for non-monotonic associations. All the models were controlled by the infant’s age, BMI z-score, and sex, and maternal IQ score. Because the relationship between the outcome and age may not be linear, squared age was included in the models when it was significant. The statistical analyses were carried out using the statistical package STATA 14 [28].

### 2.5. Spatial Analysis

An exploratory analysis was conducted on spatial data to identify possible aggregations of subjects with higher or lower neurodevelopment scores or blood metal concentrations in certain parts of the study area, and the bivariate Moran’s I index was used to test and visualize the spatial correlations between neurodevelopment scores and blood metal concentrations using ArcGis [29]. This could help identify potential environmental sources of exposure that participants did not recognize. It was also used to determine whether participants were more likely to abandon the study depending on their locality of origin, given that the health clinic could be less accessible.

Participants’ homes were each geographically referenced with a global positioning system (GPS) eTrex^®^ Legend (Garmin Ltd., Olathe, KS, USA) with an accuracy of ±3 m. Once each home was georeferenced and studied within each municipality, it was subdivided as urban or rural, except for Navojoa, whose population was solely rural, according to the AGEB classification (Spanish acronym for Basic Geostatistical Area) as determined by the Mexican National Institute of Statistics and Geography (INEGI) [30].

## 3. Results

The participants came from 33 localities belonging to the following four municipalities of the State of Sonora in northern Mexico: Huatabampo (48%), Benito Juárez (22%), Etchojoa (19%), and Navojoa (12%). Maternal mean (SD) blood levels (µg/L) of the studied metals during the third trimester of pregnancy were: Pb = 11.20 (0.97), Hg = 2.10 (1.0), and Mn = 10.26 (3.4). The mean maternal IQ was borderline: 73 points. Nonetheless, the distribution of maternal IQ was random in the study area. Blood Pb, Hg, or Mn levels were also randomly distributed in the study area. The value of global Moran’s I indicated a non-significantly global spatial autocorrelation of neurodevelopment scores and blood metal concentrations across the entire study area. The participants lost to follow-up did not show a geographical distribution pattern.

Women whose infants did not complete at least one BSID-III evaluation, and hence were not included in the study (n = 47), showed no significant differences in their mean WASI score (73 points) or blood metal concentrations (Pb = 11.05 µg/L, Hg = 1.89 µg/L, and Mn = 9.33 µg/L). However, women whose infants completed the four BSID-III evaluations did have higher median WASI scores (78 vs. 72) and pre-pregnancy BMI (28 vs. 26) than women whose infants lacked at least one evaluation (borderline significance); no other significant differences were seen among them.

A description of the study subjects’ main characteristics and the comparison between those who completed all tests and those who did not is shown in Table 1.

Mean language, cognitive, and motor development indices of the offspring at each age of evaluation were between low-average to average: 84.5–102.6 points. We observed an apparent declining tendency in the language scale as age increased during the first year of life, but not in the motor and cognitive scales; however, when a Mann–Kendall trend test was performed, we found no statistical differences by age in any scale (*p* > 0.05) (Figure 2).

Maternal blood Pb levels were negatively associated with the offspring’s language, cognitive, and motor development; however, only the association with language development was statistically significant (*p* < 0.05). Mercury blood levels were negatively associated with language and cognitive development, but neither association was statistically significant. Manganese blood levels in the third tertile were negatively associated with all three neurodevelopment indices, whereas those in the first tertile were only negatively associated with language and motor development; no index and Mn association was statistically significant (Table 2).

Multilevel model results showed that language development coefficients of the child decreased by 1.5 points per 1 µg/dL increase in blood Pb levels (*p* = 0.002). This association was stronger in children with blood Mn levels < 9.6 µg/L (ß = −1.9, *p* = 0.003) or if children had blood Hg levels > 1.9 µg/L (ß = −1.6, *p* = 0.013). Motor development had a negative and statistically significant association with Pb blood (ß = −1.06, *p* = 0.019) when the interaction term between Pb, Mn, and Hg was included in the model (ß = −0.037, *p* = 0.03). Even though the interactions between Pb and Hg, and Pb and Mn were positive and showed a statistical significance of *p* < 0.05 for the motor scale, the individual coefficient of each metal was negative and non-significant; thus, the net effect was irrelevant. No other interactions were found to be significant in the rest of the models (Table 3).

## 4. Discussion

The negative associations seen here between blood Pb levels and neurodevelopment suggest that prenatal exposure to Pb in concentrations as low as 1–5 µg/dL may affect neurodevelopment, especially language development, as has been suggested previously [31]. The negative effects of these low Pb concentrations seem to be increased by Mn deficiency of co-exposure to low levels of Hg. Whether the associations found in this study stand as the child turns older is beyond the scope of this study.

In 99% of the participants, blood Pb levels were below Mexico’s current 5 µg/dL reference value [32]; this is congruent with the infrequent use of lead-glazed ceramics (consistently found as the principal lead exposure source in Mexico’s general population after lead was phased out from gasoline in 1998) reported in the questionnaire and lower than the national Pb poisoning prevalence of children between one and four years old in Sonora, according to Mexico’s National Health and Nutrition Survey [33]. Additionally, no other important Pb sources were identified in the study area.

Even though Mn deficiency is not common, the levels found in this study were much lower than those seen in other studies. These levels might be partially explained by a possible steep decline in Mn levels in the prenatal period, which has been reported in dentine [34]. Complementing the previous explanation, when we compare maternal blood Mn levels with those of other studies that measure cord blood levels, we must take into account that since Mn is an essential element, it is actively transported through the placenta to the fetus [12]. Therefore, Mn levels in cord blood are higher than in maternal blood, which is what we measured [35].

Since diet is the main source of Mn uptake in the general population, especially through whole grains, nuts, leafy vegetables, and teas, it would not be unlikely to see Mn deficiencies in this population characterized by low income and poor education. Since micronutrient deficiencies seldom occur alone, low Mn levels could also be a proxy for other more common micronutrient deficiencies that are also associated with intellectual impairments, such as iron; however, we did not measure hemoglobin levels. Notwithstanding, the length-for-age of these children did not indicate stunting [36], which leads us to assume they were not severely undernourished.

The relatively low blood metal levels found and the fact that no geographical aggregations of high levels of Pb, Hg, or Mn were seen suggests that there were no unidentified sources of environmental exposure to these metals by themselves or in combinations. It should be noted that the maternal blood samples were all obtained before the 6 August 2014 chemical spill from a copper mine into the Bacanuchi and Sonora rivers [37].

Manganese’s potential to act both as an essential element and as a toxicant has been evidenced by the inverted U-shaped associations reported between blood Mn levels measured at delivery or 12 months of age and BSID-III mental development scores at 6–12 months of age. The levels at which maternal-blood Mn transitions from beneficial to toxic for the offspring’s six-month mental development scores were 24–28 μg/L [38,39]. Our study’s prenatal levels of Mn were not high enough to explore cut-off points seen in previous studies; prenatal Mn levels in the highest tertile of our study participants were below the 24–28 μg/L threshold. Hence, we did not find an inverted U-shaped association between maternal Mn blood levels and neurodevelopment indices.

The adverse neurodevelopmental effect of prenatal co-exposure to Pb and Mn has been studied in scenarios where Mn levels have been high. When adjusted cord blood Mn (ranging from 17.9 to 106.9 μg/L) and Pb (ranging from 0.16 to 43.2 μg/L) surpass the 75th percentile, they have a negative and statistically significant association with overall, cognitive, and language quotients of the Comprehensive Developmental Inventory for Infants and Toddlers (CDIIT) in two-year-old children from Taiwan [40]. Additionally, a cohort study of 455 children showed a significant interaction between Pb and Mn over time, suggesting that Pb toxicity is higher among children with elevated Mn co-exposure. Negative and statistically significant associations were found between the highest Mn quintile × continuous Pb and parameters of the BSID-III: mental development score, β = −1.3 (95% CI −2.2 to −0.4); psychomotor development score, β= −0.9 (95% CI −1.8 to −0.1). The coefficients for the estimated 12-month Pb effect on 18-month mental development and 24–36-month psychomotor development scores were larger for children with high Mn compared with those of children having midrange Mn levels [39]. However, we did not find studies reporting increased toxicity of Pb in the presence of low Mn levels, as we saw in this study; this might be the first study to do so.

Maybe the low manganese levels found in this population could reflect a poor diet in general and more specifically, a diet deficient in nutrients that protect against lead absorption and adverse effects (calcium, iron, zinc, and vitamin D) [41], thus explaining the enhanced adverse effect of lead in the lowest Mn tertile. However, we have no way of proving this hypothesis. We also have no knowledge of optimal manganese blood concentrations during the third trimester of pregnancy for fetal neurodevelopment to conclude on the possible direct effects of the concentrations found in this study. Since we did not have dietary information on the participants, we were unable to conclude on interaction with vitamins, minerals, and other dietary factors that interfere with metal toxicokinetics; it is up to future studies to elucidate this matter.

Despite Sonora being a coastal state, and therefore fish presumably an important component of the local diet, Hg levels were not high. Almost 90% of the pregnant women in this study were below the threshold for adverse neurodevelopmental effects of 3.4 µg/L in maternal blood, estimated using the mean ratio of 1.7 cord blood Hg–1 maternal blood Hg [42] and the threshold of 5.8 μg/L cord blood Hg for adverse effects [43]. Therefore, it was not surprising that we did not find a negative association between low prenatal levels of Hg and neurodevelopment when it was studied by itself. In similar situations, low levels of Hg have not been associated with a detrimental effect or have even been associated with positive effects in neurodevelopment, perhaps due to the benefits of maternal fish consumption during pregnancy, which is commonly the primary source of exposure to Hg [44,45]. The interesting observation is that these low prenatal blood Hg levels might also potentiate the effects of low prenatal blood Pb levels. A similar effect was observed in a study using the Korean version of the Bayley scale of infant and toddler development-II, where those above the 50th percentile of late prenatal Hg potentiated the negative effect of Pb on mental development index (β = −4.33 (−7.66, −1.00)) [46].

The fact that motor development in our study showed a negative association with blood Pb concentrations but the statistical significance was marginal suggests there might be an effect at this level, but we did not have enough power to detect it.

Cognitive neurodevelopment was not associated with any of the metals in this study. Perhaps the threshold for cognitive effects is higher than for language and motor ones. This is speculated based on the fact that a significant association was seen in another study [40], but their manganese concentrations were much higher than what we found in our study (17.88–106.85 µg/L, mean = 47.90 µg/L).

The main limitations of our study were the possibility of residual confounding and a possible lack of power to detect motor and cognitive associations. Since nutritional status was not assessed (we lacked hemoglobin measurements and food frequency questionnaires), we cannot rule out that associations seen for Mn in fact arose from that element or other unmeasured nutrient. In addition, Hg measurements were not speciated, so we had no methyl mercury measurements to compare with other studies. Finally, we did not measure other potentially present neurotoxicants, such as arsenic.

Our results contribute to a better understanding of real-life susceptibility factors regarding the negative association of metals and neurodevelopment: timing, co-exposures, and perhaps nutritional circumstances [47]. We especially provide new and relevant data on Mn’s neurotoxic effects and interactions in offspring through its deficiency during pregnancy, but more studies will be needed to further elucidate the role of these modifiable factors.

## 5. Conclusions

Prenatal exposure to low Pb levels, resulting in maternal blood levels well below the current Mexico and US reference values (5 and 3.5 µg/dL, respectively), may impair the offspring’s neurodevelopment during the first year of life. The effect is especially evident in language development, and even more so if the pregnant women are also exposed to low levels of Hg, or apparently, if they are deficient in essential nutrients such as Mn.

This is important considering that most of the burden of disease of Pb, Hg, and Mn will be in scenarios of low-level exposures. The co-existence in pregnant women of micronutrient deficiency and exposure to low background levels of Pb and Hg is not uncommon, especially in developing countries. However, these conditions are preventable, albeit especially challenging where they are most prevalent. Given the associations we observed at relatively low Hg and Pb blood levels and low average IQ levels, it is of paramount importance to prevent malnutrition and exposure to these toxic metals as much as possible to improve the population’s wellbeing. Even though the metals’ blood concentrations were low and we did not find obvious sources of exposure, the fact that our results showed significant and negative associations with the offspring’s neurodevelopment should give rise to investigating the sources in order to further reduce the exposure.

Further research is required to characterize the dose–response curve of the co-exposure associations at other levels and to investigate if the associations found hold after the first year of life.

This study highlights the importance of considering more complex, real-life scenarios of exposure to different metals that can affect neurodevelopment.

## Figures and Tables

**Figure 1 ijerph-19-13020-f001:**
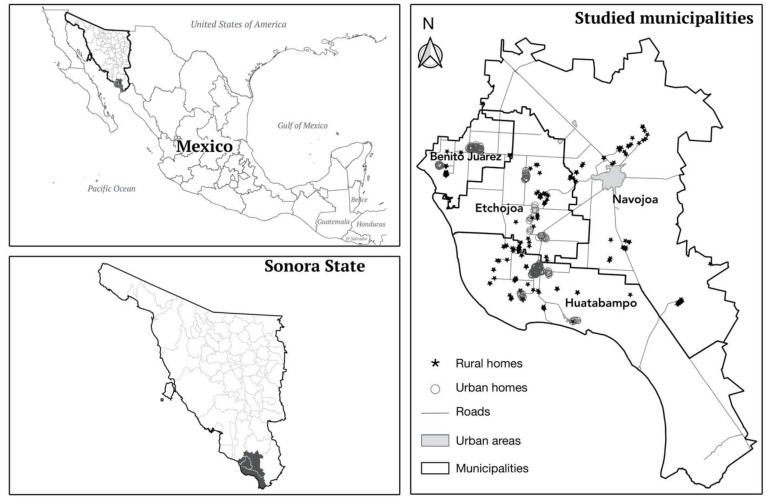
Study site.

**Figure 2 ijerph-19-13020-f002:**
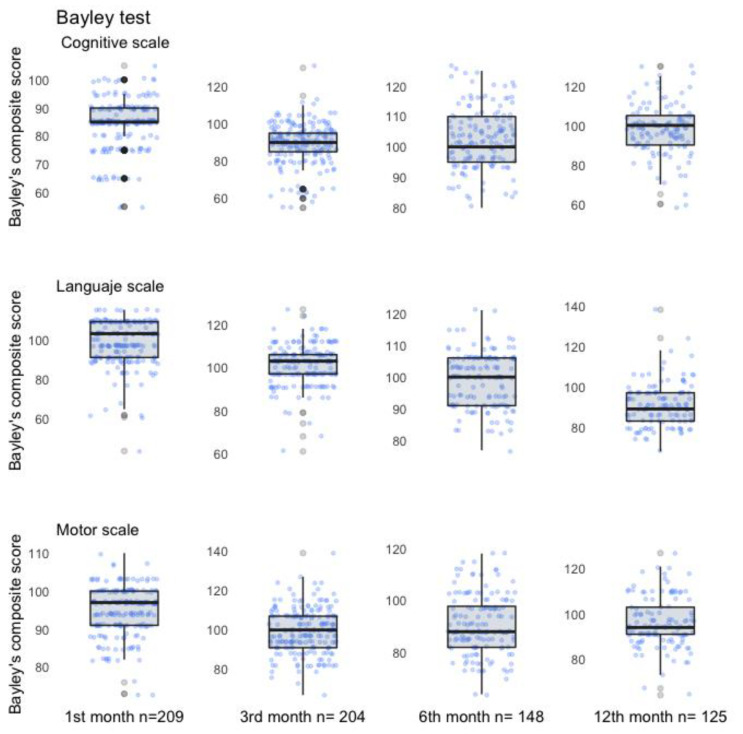
Bayley Scale III. Scores subscales by age.

**Table 1 ijerph-19-13020-t001:** Study subjects’ main characteristics: comparison of medians and interquartile ranges according to completion of study evaluations.

Characteristic	All ^a^	Complete Information ^b^	Incomplete Information ^c^	*p*-Value
	Q_50_ (Q_25_–Q_75_)	Q_50_ (Q_25_–Q_75_)	Q_50_ (Q_25_–Q_75_)	
Mother				
Blood Pb (µg/L)	7.60 (5.00–14.30)	7.80 (5.65–14.70)	7.55 (4.60–14.25)	0.83
Blood Hg (µg/L)	1.93 (1.34–2.77)	1.79 (1.25–2.35)	2.01 (13.40–2.80)	0.49
Blood Mn (µg/L)	9.92 (7.65–12.45)	9.07 (7.51–11.80)	10.15 (7.56–12.95)	0.22
Age (years)	23 (20–27)	21 (19–26)	23 (20–27)	0.09
IQ score (points)	73 (64- 84)	78 (68–93)	72 (64–83)	0.05
Pre-pregnancy BMI	26.34 (23.12–29.07)	28.24 (24.34–31.25)	26.10 (23.05–28.46)	0.05
Offspring				
Birth weight (kg)	3.40 (3.15–3.70)	3.40 (3.20–3.75)	3.40 (3.10–3.67)	0.77
Birth length (cm)	53.00 (51.00–55.00)	54.00 (52.00–55.00)	53.00 (51.00–55.00)	0.30
Head circumference (cm)	36 (35.00–36.70)	35.50 (34.20–36.50)	36.00 (35.00–36.70)	0.25
Vaginal Birth (%)	71.3	65.9	72.5	0.38
Female sex (%)	47.9	47.3	48.1	0.91

^a^ All children. n = 253. ^b^ Four BSID-III evaluations in one year. n = 53. ^c^ One to three BSID-III evaluations in one year. n = 200.

**Table 2 ijerph-19-13020-t002:** Neurodevelopment association with prenatal blood lead, mercury, and manganese levels in 1–12-months-old children in Sonora, Mexico ^a^.

Metals (µg/L)	Bayley Scales		
	Language	Motor	Cognitive
	ß (CI 95%)	ß (CI 95%)	ß (CI 95%)
	n = 522		
Hg ^b^	−0.51 (−1.43–0.41)	0.04 (−0.89–0.97)	−0.25 (−1.16–0.67)
Pb ^c^	−0.15 (−0.24–−0.56) **	−0.08 (−0.17–0.01)	−0.01 (−0.10–0.08)
	n = 252		
Pb only in women with Hg >1.9 ^b^	−0.16 (−0.29–−0.34) **	−0.08 (−0.20–0.04)	−0.03 (−0.15–0.09)
	n = 247		
Pb only in women with Mn <9.6 ^b^	−0.19 (−0.31–−0.06) **	−0.01 (−0.02–0.02)	−0.08 (−0.02–0.04)
Mn ^b^	n = 364		
1st tertile (2.40–6.86)	−1.21 (−3.71–1.30)	−1.02 (−3.55–1.52)	0.21 (−2.16–2.58)
2nd tertile (6.87–12.80)	Ref.	Ref.	Ref.
3rd tertile (12.9–21.30)	−1.95 (−4.30–0.40)	−1.20 (−3.58–1.17)	−1.25 (−3.48–0.97)

^a^ Associations evaluated through multilevel models based on child and municipality of origin. ^b^ Model adjusted by offspring’s sex, BMI (z—score), age, and maternal IQ. ^c^ Model adjusted by offspring’s sex, BMI (z—score), age (age + age2), and maternal IQ. ** *p* < 0.01.

**Table 3 ijerph-19-13020-t003:** Joint association of blood lead, mercury, and manganese with language, motor, and cognitive development in children 1–12 months of age ^a^.

Metals (µg/L)	Bayley Scales		
	Language	Motor	Cognitive
	ß (CI 95%)	ß (CI 95%)	ß (CI 95%)
Pb	−0.35 (−0.12–0.52)	−1.06 (−1.95–−0.17) **	−0.13 (−0.97–0.70)
Hg	0.07 (−4.73–4.86)	−2.79 (−7.70–2.11)	0.68 (−3.98–5.34)
Mn	−0.30 (−1.32–0.73)	−0.82 (−1.87–0.22)	0.09 (−0.90–1.08)
Pb # Hg	0.04 (−0.32–0.39)	0.38 (0.02–0.75) *	0.03 (−0.32–0.37)
Pb # Mn	0.03 (−0.05–0.11)	0.09 (0.01–0.18) *	0.004 (−0.74–0.83)
Hg # Mn	0.006 (−0.44–0.45)	0.29 (−0.17–0.75)	−0.11 (−0.31–0.33)
Pb # Hg # Mn	0.007 (−0.04–0.03)	−0.04 (−0.07–−0.003) *	0.0008 (−0.03–0.03)

n = 522 observations. ^a^ Multilevel models based on municipality subdivision and child and adjusted by age, maternal IQ, child’s age, sex, and BMI (z-score). * *p* < 0.05; ** *p* < 0.01.

## Data Availability

The data presented in this study are available on request from the corresponding author. The data are not publicly available due to privacy issues.

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
