# Peer review of "Prenatal Co-Exposure to Manganese, Mercury, and Lead, and Neurodevelopment in Children during the First Year of Life"

_ijerph, 2022, doi:10.3390/ijerph192013020_

Round 1

Reviewer 1 Report

Although this study has many significant research results, it has illustrated the correlation between metal exposure and infant neurodevelopment. However, many studies on this topic have been published in the past, and the results are less innovative. In addition, there are many omissions and inadequacies in the description of materials and methods, as well as the presentation of results, and the overall article should be significantly revised.

Introduction:

1.      In Line 39-40, it mentioned the environmental pollutants are not blocked by placenta or BBB. It suggested to add some reference to focus on abovementioned mechanisms for Mn, Hg, and Pb.

Materials and Methods:

1.      In Line 77, it is better to compare the difference between rural and suburban study subjects.

2.      Please give the full name of APGAR in Line 92.

3.      The symbol in Figure 1 is not clear, and it should be enlarged and clearer.

4.      In Line 97, the number of approved IRB should be added.

5.      For 2.2.2 Neurodevelopment assessment, the timing for data collection of BSID-III by infants should be mentioned in this section.

6.      For 2.3.1-3, it suggested to combined the QA/QC procedure and result with more detailed description.

7.      For 2.4 Statistical analysis, the first section mentioned the exploratory analysis to observe the shape and distribution of variables, but there was no description to illustrate the next step to deal with these variables. In addition, it seems to applied the regression model, and there was no related description.

8.      For 2.5 Spatial analysis, I did not figure out obvious result conducted by it.

Results:

1.      As to Table 1, it should be added the data for total study subjects and subjected mentioned in Line 202 to comprehensive illustrate the whole picture for the study subjects.

2.      There was no any description to present the Figure 2.

3.      For Table 2 and 3, it has mentioned the Bayley’s scales collected by different time of the first year of infants. However, I can not figure out how to use or adjust these time-dependent data in these results.

4.      As to Table 2, it is confused for the definition of age and age+age2. In addition, how to decide the value of 1.9 for Hg and 9.6 for Mn. And, why the Hg and Pb conducted by continuous data and Mn by tertile.

5.      In Line 237, it cannot be found out the data from any Table or figure.

6.      As to Table 3, some significant results shown in this table did not be described in the main text.

Discussion:

1.      In Line 250-251, how to define the Mn deficiency in the present study? And, it would be all Mn deficiency of study subjects depending on the evidence in Line 285-287, and the analysis and grouping for Mn in Table 2 should be modified.

2.      It should add one section to illustrate the limitation of this study.

Reviewer 2 Report

This study aimed to evaluate the possible associations of Pb, Hg, and Mn prenatal levels (jointly and separately) with neurodevelopment in the first year of life.

This is a scientifically well-conducted and well-written work whose implications could have an interesting clinical implication and the development of other literature on the subject. 

There is plagiarism to be corrected.

Minor Points:

INTRODUCTION: The authors should clarify what are the situations (diet, environment or other) in which simultaneous contact with multiple combinations of different levels of Pb, Hg, and Mn could produce synergistic or antagonistic effects.

METHODS It is not clear why the subjects participant's home was geographically referenced using a GPS.

DISCUSSION The authors should clarify what the stratgies might be to reduce the risk of contamination by Pb and Hg in this subjects.

Round 2

Reviewer 2 Report

The authors fixed the paer following my observations. There is still some plagiarism, especially among the methods. 
